# Factors Influencing Self-Confidence and Educational Needs in Electrocardiographic Monitoring Among Emergency Room and Intensive Care Unit Nurses

**DOI:** 10.3390/healthcare13030277

**Published:** 2025-01-30

**Authors:** Miji Kim, Jaeyong Yoo

**Affiliations:** 1Department of Nursing, Chosun University Hospital, Gwangju 61453, Republic of Korea; 2010087@csuh.co.kr; 2Department of Nursing, College of Medicine, Chosun University, Gwangju 61452, Republic of Korea

**Keywords:** electrocardiogram monitoring, self-confidence, nurses, cardiac care, emergency room, intensive care units, educational needs

## Abstract

The self-confidence of nurses in performing electrocardiographic (ECG) monitoring is a critical competency for managing patients with cardiac conditions in high-acuity settings such as emergency rooms (ERs) and intensive care units (ICUs). This study aimed to identify the factors influencing nurses’ confidence in ECG monitoring and to assess their educational needs. A total of 153 ER and ICU nurses participated in this cross-sectional study, completing structured questionnaires assessing their knowledge, confidence, and educational needs regarding ECG monitoring. The findings revealed a moderate mean confidence score of 63.47 (±15.09) out of 100. The key factors associated with higher confidence included the completion of ECG-related education, frequency of evidence searching, and clinical experience within the current department. Additionally, nurses prioritized eight critical educational topics for improving ECG-monitoring competency. These results underscore the importance of tailored educational programs and systematic training strategies to address identified gaps in knowledge and confidence. By prioritizing the specific needs of ER and ICU nurses, healthcare systems can foster supportive work environments, enhance nursing practice, and ultimately improve patient outcomes. Future research should evaluate the long-term impact of educational interventions on nurses’ performance and clinical outcomes.

## 1. Introduction

With the aging population, the incidence of heart disease and sudden cardiac arrest is on the rise [1]. In the United States, more than 292,000 in-hospital cardiac arrests occur annually, with 50–60% of these cases being attributed to heart-related conditions [2]. The incidence of acute cardiac arrest in Korea has increased by approximately 60% over the past decade, highlighting the growing severity of cardiac emergencies [3]. Despite advancements in medical technology, the survival discharge rate after in-hospital cardiac arrest remains low at 8.4% [4]. Emergency events such as abnormalities in electrocardiogram (ECG) waveforms, ventricular fibrillation, and pulseless ventricular tachycardia necessitate immediate intervention, as delay in treatment can lead to life-threatening consequences [5].

Most emergency situations in hospitals are witnessed by medical staff [6], with nurses being particularly positioned as the first to recognize changes in patients’ conditions and frequently observing emergency events [7]. Early ECG findings significantly impact the rate of recovery of spontaneous circulation and survival [5]. Nurses play a crucial role in saving patients’ lives by promptly interpreting and responding to these findings [8,9]. In particular, emergency room (ER) and intensive care unit (ICU) nurses closely monitor ECGs and provide care based on this vital information [10]. ECG monitoring is a nursing process that involves the continuous recording and assessment of the patient’s ECG through the use of equipment [11]. This includes tasks such as proper electrode placement, setting monitoring goals, selecting leads and alarm parameters, detecting changes through monitoring, and implementing appropriate interventions and evaluations [12]. The accurate interpretation of ECGs and timely responses through ECG monitoring are essential competencies for ER and ICU nurses, as they help reduce errors in emergency situations and ensure patient safety [12,13]. Strengthening these competencies not only improves treatment outcomes but also enhances nurses’ confidence in managing cardiac emergencies [14,15]. Effective ECG monitoring enables the early detection of heart problems, facilitating timely interventions that can prevent fatal events and improve patient outcomes [5,6,10,12]. Therefore, possessing accurate knowledge and confidence in ECG monitoring is critical for nurses [13,16].

Confidence in performing clinical tasks, such as ECG monitoring, is closely aligned with Bandura’s Self-Efficacy Theory, which emphasizes that individuals’ belief in their ability to successfully execute a task directly influences their performance [17]. According to this theory, self-efficacy is derived from four key sources: mastery experiences, which involve hands-on practice and successful task execution; vicarious experiences, gained through observing peers or mentors perform tasks effectively; verbal persuasion, including encouragement and feedback from educators or colleagues; and physiological and emotional states, such as stress levels and physical readiness, which can influence confidence [17]. Within the context of nursing, self-efficacy is particularly critical for effective decision making and accurate task execution in high-acuity environments like the ER and ICU [18,19,20]. Grounded in the Self-Efficacy Theory, our study investigates how nurses’ knowledge, clinical experience, and participation in targeted educational interventions might enhance their confidence in performing ECG monitoring. Incorporating the Self-Efficacy Theory provides a robust lens through which to interpret the factors influencing nurses’ confidence, supporting efforts to improve both educational strategies and clinical outcomes.

Previous studies highlight that nurses’ confidence in performing ECG monitoring is significantly influenced by three key factors: knowledge, experience, and educational background [13,21,22,23]. For example, higher levels of theoretical knowledge have been linked to improved confidence in rhythm interpretation and clinical decision making [13,22]. Similarly, hands-on experience in cardiac care settings fosters practical competency, further boosting confidence levels [21]. Educational interventions, such as simulation-based training or refresher courses, have also been shown to enhance nurses’ performance confidence in ECG-related tasks [23]. These challenges are not unique to Korea but align with findings from international studies, which also highlight the need for structured training and continuous professional development to address knowledge and confidence gaps in ECG monitoring [10,12,13]. Despite these findings, research specifically examining the confidence and knowledge of ER and ICU nurses—who manage high-acuity cardiac patients—is limited. This underscores the need for targeted investigations addressing their unique educational and clinical demands in ECG monitoring.

ECG monitoring is one of the twenty core basic nursing skills identified by the Korea Board of Nursing Education and Evaluation (KABONE) and is introduced at the undergraduate level [24]. However, due to a lack of formal educational courses and professional development programs in clinical settings, nurses face challenges in acquiring the necessary knowledge, skills, performance confidence, and competence in ECG monitoring [7,25,26]. Therefore, it is crucial to establish an ECG monitoring education and training program for ER and ICU nurses who frequently manage patients with heart-related conditions. This study aims to assess the knowledge and performance confidence of these nurses regarding ECG monitoring and to identify the factors that may enhance their confidence. By analyzing educational needs, the study seeks to provide foundational data for developing consumer-centered educational programs.

## 2. Materials and Methods

### 2.1. Study Design and Data Collection

This study employed a descriptive cross-sectional research design to identify the factors influencing ER and ICU nurses’ performance confidence in ECG monitoring and to assess their educational needs related to ECG monitoring. Data collection was carried out between September and October 2021. Following approval for research collaboration from the nursing administrator of C University Hospital, the researcher visited the ER and ICU departments to present the study to the department managers. The researcher provided a detailed explanation of the study’s purpose and methodology to potential participants, obtained written informed consent, and emphasized the importance of honest and accurate responses. The questionnaire was self-administered and required approximately 15 to 20 min to complete. Upon completion, the participants were instructed to seal their completed questionnaires in the provided return envelopes and deposit them in a designated return box located within their departments.

### 2.2. Study Participants and Setting

This study was conducted with nurses working in one ER and four ICUs at C University-affiliated tertiary Hospital, located in G Metropolitan City, Korea. The study participants were ER and ICU nurses who understood the purpose and procedures of the study and voluntarily provided written consent. However, new nurses with less than 3 months of work experience and ward managers who did not directly provide patient care were excluded. The sample size for this study was calculated using the G*Power 3.1.9 program, with an effect size of 0.15 derived from previous studies on ECG-related education and confidence among clinical nurses [27,28]. This effect size was selected to detect meaningful relationships in studies that examine multifactorial influences on confidence and knowledge, ensuring sufficient statistical power (80%) and a significance level of *p* < 0.05. This approach aligns with prior research that highlights the appropriateness of medium effect sizes for exploring similar constructs in education and clinical confidence. Incorporating five predictors into the analysis, the required sample size was determined to be 138 participants. Considering a 20% dropout rate, 166 participants were selected. After excluding 13 cases with insincere responses, the final analysis included 153 subjects. The demographic and professional characteristics of ER and ICU nurses included gender, age, department, education level, professional position, work type, total clinical experience, clinical experience within their current department, and the average number of cardiac patients managed in the past week. Furthermore, factors such as the frequency of evidence-based practices related to ECG monitoring, completion of relevant educational programs, participation in research activities, and possession of professional certifications were also analyzed.

### 2.3. Measurements

This study utilized a structured self-report questionnaire related to ECG monitoring. The use of this tool was approved in advance by the developers of the questionnaire. To ensure the clinical relevance of the questionnaires, the existing validated tools were reviewed, focusing on ECG monitoring tasks specific to ER and ICU settings. A panel of clinical experts evaluated content validity, and a pilot test with 10 ER and ICU nurses provided feedback on clarity and relevance. Minor revisions were made based on this feedback to ensure the questionnaire addressed the unique demands of ECG monitoring in high-acuity environments.

#### 2.3.1. ECG Monitoring Knowledge

Knowledge of ECG monitoring was assessed using the measurement tool developed by Yeom et al. [29]. This tool, based on the Korean Nurse Licensure Examination format, consists of 20 questions: 4 questions related to the characteristics of normal ECG waves, 9 questions related to abnormal wave characteristics, and 7 questions related to the visual representation of ECG waveforms. Each correct answer is scored as 1 point, while incorrect answers receive 0 points, with a higher score indicating greater ECG monitoring knowledge. Examples of questions include “What are the ECG characteristics that may occur in a patient who has received a large dose of digoxin?” and “Which of the following statements about premature ventricular contractions is correct?” In this study, a pilot test was conducted with 10 clinical nurses, each with 3 to 10 years of experience in the ER and ICU, to evaluate the appropriateness and clarity of the question structure and expressions. The content validity index (CVI) for all the questions was evaluated as appropriate, with a score of 0.80 or higher, and the tool was used without modification. At the time of tool development, the reliability coefficient, measured by the Kuder–Richardson 20 (KR 20) value, was 0.83. In this study, the KR20 value was 0.74, indicating acceptable internal consistency.

#### 2.3.2. Confidence in Performing ECG Monitoring

To evaluate performance confidence in ECG monitoring, this study utilized a validated instrument developed by Yeom et al. [29]. The tool comprises 10 items, of which six items assess confidence in understanding basic ECG concepts (maximum score: 60 points), and the remaining four items measure confidence in applying ECG monitoring in clinical practice (maximum score: 40 points). The overall confidence score is calculated as the sum of these two subcomponent scores, yielding a total score ranging from 0 to 100, with higher scores indicating greater confidence in performing ECG monitoring. For example, if a participant scores 50 points in understanding ECG concepts and 30 points in applying ECG in clinical practice, their total confidence score will be 80 out of 100. This scoring method ensures that both theoretical understanding and practical application are equally represented in the overall confidence score. Examples of confidence questions include “I can calculate the patient’s heart rate through an electrocardiogram recorder.” and “I can appropriately record the nursing interventions performed and the patient’s responses”. The reliability of the tool was reported as Cronbach’s α = 0.93 at the time of its development, and Cronbach’s α in this study was 0.92.

#### 2.3.3. ECG Monitoring Educational Needs and Priorities

The research team developed an educational needs and priorities assessment for ECG monitoring based on the practice standards set out by the American Heart Association (AHA) for ECG monitoring [30]. The assessment included 5 categories: need for ECG monitoring-related education; willingness to participate; preferred educational methods; education time and frequency; and the importance and urgency of education topics. Educational topics consisted of 35 detailed items based on recommendations provided by the AHA [30]. To evaluate the content validity of the detailed items, the CVI was assessed by two emergency physicians, one ICU and ER nurse with over 10 years of experience, and one nursing professor. Only items with a CVI of 0.80 or higher were selected, resulting in the removal of 15 items deemed unnecessary and the confirmation of a final set of 20 detailed items. To identify priority educational topics, the participants rated each item on a scale from 0 to 10 for both importance and urgency. These ratings were analyzed using the Eisenhower matrix, a decision-making tool traditionally used to prioritize tasks by assessing their importance and urgency. Using Importance–Performance Analysis (IPA), the mean values for importance and urgency were calculated for each topic and plotted onto a four-quadrant matrix: Quadrant 1 (Important and Urgent), Quadrant 2 (Important but Not Urgent), Quadrant 3 (Urgent but Not Important), and Quadrant 4 (Neither Important Nor Urgent). This structured approach ensured that educational needs were aligned with practical relevance and urgency, allowing limited resources to be allocated to the most impactful topics.

### 2.4. Data Analysis

Data were analyzed using SPSS/WIN 26.0, with statistical significance set at *p* < 0.05. A Shapiro–Wilk test confirmed that the main study variables followed a normal distribution, justifying the use of parametric statistical analyses. Descriptive statistics were employed to summarize the participants’ demographic characteristics, ECG monitoring knowledge, and performance confidence. Group comparisons were conducted using independent *t*-tests and ANOVA to examine the differences in confidence levels based on characteristics such as clinical experience, the completion of ECG-related education, and frequency of evidence searches. Pearson’s correlation coefficient was used to assess relationships between key variables, including ECG monitoring knowledge and confidence.

To determine the factors influencing confidence in ECG monitoring, a multiple regression analysis was performed with confidence as the dependent variable. The independent variables included the completion of ECG-related education, frequency of evidence searches, and clinical experience within the current department. Assumptions for regression analysis were tested, confirming linearity, homoscedasticity, and the absence of autocorrelation (Durbin–Watson: 1.917–1.972). Multicollinearity was also ruled out, with tolerance values ranging from 0.925 to 0.976 and VIF values from 1.025 to 1.081. One outlier was excluded to maintain normality. Educational needs were assessed using the Eisenhower matrix, which prioritized topics based on importance and urgency. Topics categorized as Important and Urgent (Quadrant 1) were identified as priorities for targeted educational interventions. This analytical approach ensured reliable results aligned with the study’s objective to identify factors and educational needs influencing ECG monitoring confidence.

### 2.5. Ethical Considerations

This study received ethical approval from the Institutional Review Board (IRB) of C University Hospital (Approval Number: 2021-07-031-003). Prior to data collection, the participants were provided with detailed information regarding the purpose, content, and expected time commitment of the study through a formal explanation session. Written informed consent was obtained from all the participants. The participants were informed of their right to withdraw from the study at any time without repercussions, and the measures taken to ensure the anonymity and confidentiality of their information were explained in detail. It was explicitly stated that all the data collected would be used for research purposes only and would be securely stored in an encrypted computer system for three years in accordance with institutional and regulatory guidelines, after which it would be permanently deleted. As a token of appreciation, participants who completed the survey received a small gift valued at USD 5.

## 3. Results

### 3.1. General and Clinical Characteristics of Study Subjects

The study participants were 94.1% female nurses with an average age of 29.7 years. In terms of educational level, 88.2% had a bachelor’s degree, while 11.8% had a master’s degree or higher. The majority (93.5%) worked in a three-shift system. The mean total clinical experience was 78.32 months, with a mean of 38.37 months in the current department. On average, the nurses cared for 6.36 cardiac patients per week (Table 1).

Regarding ECG monitoring resources, senior nurses (59.5%) and fellow nurses (54.3%) were the most frequently consulted, followed by physicians (47.7%) and textbooks (43.8%). Less commonly utilized resources included librarians (72.5% never used), theses or journals (66.0% never used), and conference materials (69.3% rarely or never used). When searching for ECG-related evidence, 38.6% conducted searches “once every few months,” relying mostly on general web engines and ward manuals. However, 52.9%, 69.9%, and 71.9% reported not using Medline, PubMed, or CINAHL, respectively.

Fifty-eight percent (58.2%) had never received formal ECG monitoring training. Among those with training, most (57.8%) attended external programs, with an average of 1.47 sessions. Over half (54.7%) had last attended training more than two years prior. More than two-thirds (72.5%) held Basic Life Support (BLS) certification. Only 2.6% had ever participated in a clinical study on ECG monitoring, and 0.7% had attended a relevant conference in the past year (Table 1).

### 3.2. ECG Monitoring Knowledge and Performance Confidence

The participants scored an average of 13.01 (SD ± 3.08) out of 20 on the ECG knowledge test. Sub-component analysis revealed that the participants achieved a mean score of 2.86 (SD ± 1.07) out of 4 for knowledge of normal waveforms, 5.60 (SD ± 1.66) out of 9 for abnormal wave characteristics, and 4.54 (SD ± 1.29) out of 7 for waveform interpretation. The highest percentages of correct responses were observed in recognizing ventricular fibrillation (97.4%), identifying a normal ECG rhythm (90.2%), and determining the initial treatment for ventricular fibrillation (86.3%). In contrast, the lowest accuracy was reported for the items related to third-degree atrioventricular block (9.2% correct), digoxin overdose (23.5% correct), and conduction disorders (49.7% correct).

The mean performance confidence in ECG monitoring was 63.47 (SD ± 15.09) out of 100. The subcomponent analysis showed that confidence in understanding ECG basics averaged 37.14 (SD ± 9.34) out of 60, while confidence in applying ECG monitoring to clinical practice averaged 26.32 (SD ± 6.69) out of 40. The participants reported the highest confidence in electrode placement (8.22 ± 1.94), situation judgment using ECG (6.78 ± 1.89), and arrhythmia evaluation (6.78 ± 1.85). Conversely, the lowest confidence levels were associated with recognizing basic waveforms (6.04 ± 2.30), identifying time intervals (5.10 ± 2.14), and calculating heart rate (5.03 ± 2.30) (Table 2).

### 3.3. Differences in Confidence in Performing ECG Monitoring According to Subject Characteristics

*T*-tests and ANOVA revealed that the total confidence in ECG monitoring differed significantly by department (F = 2.85, *p* = 0.026), clinical experience in the current department (t = −2.02, *p* = 0.045), frequency of evidence searches (*t* = −2.53, *p* = 0.012), the completion of ECG-related education (*t* = 2.30, *p* = 0.023), and the timing of education (*t* = 2.48, *p* = 0.016). Nurses with ≥3 years’ experience in the current department (65.43 ± 14.22), those who searched for evidence at least monthly (66.11 ± 14.84), had completed ECG education (66.73 ± 15.12), or updated training within the last two years reported higher confidence. However, a post hoc analysis found no significant differences among departments (Table 3).

Confidence in basic ECG understanding was higher for nurses working in certain departments (F = 3.10, *p* = 0.018), searching for evidence monthly (38.98 ± 9.36, *t* = −2.85, *p* = 0.005), or completing education within the past two years (42.52 ± 8.98, t = 2.89, *p* = 0.005). Confidence in applying ECG monitoring varied by age (F = 3.05, *p* = 0.030), total clinical experience (*t* = −2.77, *p* = 0.006), current department experience (*t* = −2.65, *p* = 0.009), and the completion of ECG-related education (*t* = 2.62, *p* = 0.010). Younger nurses (20–25 years) reported lower scores, though the post hoc analysis showed no statistically significant differences by age.

### 3.4. Correlation Between Variables Related to ECG Monitoring

Performance confidence in ECG monitoring correlated positively with the overall ECG knowledge (r = 0.20, *p* = 0.013) and most sub-domains of ECG knowledge, except visual waveform analysis (Table 4). This suggests that while increased theoretical knowledge generally aligns with higher confidence, certain aspects (e.g., waveform recognition) may require additional training or repeated clinical exposure.

### 3.5. Factors Affecting Confidence in Performing ECG Monitoring

Factors significantly influencing performance confidence in ECG monitoring included the completion of ECG-related education (β = 0.21, *p* = 0.008), frequency of searching for up-to-date evidence (β = 0.17, *p* = 0.032), and clinical experience within the current department (β = 0.18, *p* = 0.022). The model accounted for 14.5% of the variance and was statistically significant (F = 4.95, *p* < 0.001) (Table 5).

Examining the sub-domains, for “performance confidence in understanding ECG”, significant predictors were the completion of education (β = 0.17, *p* = 0.028) and frequency of information searches (β = 0.20, *p* = 0.015), with the model explaining 11.3% of the variance (F = 3.71, *p* = 0.003). In the sub-domain of “performance confidence related to applying ECG monitoring to nursing practice”, significant factors included the completion of education (β = 0.22, *p* = 0.004), clinical experience in the current department (β = 0.24, *p* = 0.003), and department (β = 0.16, *p* = 0.046). This model explained 16.3% of the variance and was statistically significant (F = 5.69, *p* < 0.001) (Table 5).

### 3.6. Educational Needs of Nurses Related to ECG Monitoring

In this study, 69.3% of the participants (n = 106) rated ECG monitoring education as highly necessary, and 56.2% (n = 86) expressed a strong intent to participate in future training. The preferred duration for educational programs was 1.99 ± 1.31 h per session, with the participants favoring an average of 1.99 ± 1.91 sessions per year. Online education (38.7%) emerged as the most preferred delivery method, followed by traditional lectures (33.5%) and simulation-based learning (20.2%).

Using the Eisenhower matrix, eight educational topics with above-average ratings for both importance and urgency were categorized as top priorities in Quadrant 1 (Figure 1). These priority topics included “interventions for patients with cardiac arrest“; “interventions for patients with tachyarrhythmia“; “ECG abnormalities in acute myocardial ischemia“; “interventions for premature complexes“; “interventions for patients using percutaneous pacemakers“; “defibrillation/cardioversion interventions“; “interventions for patients with bradyarrhythmias“, and “interventions for patients with syncope“.

## 4. Discussion

As healthcare systems face increasingly complex and high-stakes situations, the ability of ER and ICU nurses to perform ECG monitoring with confidence has become an essential competency that directly influences patient outcomes [12,31]. This study identified the completion of ECG-related education, frequency of searching for up-to-date evidence, and clinical experience within the current department as the key factors affecting nurses’ performance confidence in ECG monitoring. These findings are consistent with research highlighting the significance of completing education [32], maintaining the current knowledge [12], and engaging in quality clinical experience [21,33].

In our study, the mean ECG-monitoring confidence score was 63.47 (SD ± 15.09) out of 100, which parallels other low-to-moderate findings. Mohammed, Ahmed, and Ebraheim [34] reported a mean of 6.45 out of 10 in ER and ICU nurses’ ECG interpretation, while the Practical Use of the Latest Standards of Electrocardiography (PULSE) clinical study [12] indicated a mean of 49.2 among 3013 nurses. Severe ventricular arrhythmia recognition rates also remain below 30% in various settings [35]. Although direct comparisons are somewhat limited, our results align with Kim and Kang [36], who found a CPR confidence score of 76.1 in hospital nurses, and with Cho, Kim, and Chun [28], who reported 67.9 among dialysis-unit nurses. Ahn et al. [37], likewise, observed substantial challenges in ECG performance among emergency nurse practitioners, especially in percutaneous pacing, cardioversion, and defibrillation. Collectively, these data illustrate the difficulties nurses face in ECG-related tasks.

High-performance confidence is essential for effectively managing cardiac emergencies in ER and ICU [13,30,38]. However, our study revealed no significant effect of knowledge level on performance confidence, suggesting that the participants’ self-perceptions or measurement limitations may be involved. The nurses demonstrated high knowledge in areas such as ventricular fibrillation interpretation yet reported low confidence in understanding ECG waveforms, recognizing time intervals, and calculating heart rates. This discrepancy mirrors a systematic review by Chen et al. [13], indicating that nurses can possess high theoretical knowledge while struggling with real-time arrhythmia recognition. Comprehensive assessment tools must, thus, extend beyond fragmented knowledge measures to capture holistic readiness for complex scenarios. To address potential gaps, we included a preliminary survey of clinical nurses and a CVI evaluation. The participants had an average of 6.5 years of clinical experience and managed 6.3 cardiac patients weekly, suggesting that clinical experience may play a pivotal role in fostering confidence in ECG monitoring despite the gaps in theoretical knowledge. Future investigations should adopt standardized instruments tailored to ER and ICU nurses, facilitating more robust comparisons and evidence-based interventions.

Consistent with previous research [39,40,41,42], ECG-related education emerged as a critical factor in improving nurses’ confidence. McCarthy et al. [39] identified better ECG-monitoring performance among those who received specific training, whereas Tai et al. [40] reported that a Hong Kong ER nurse training program led to higher performance confidence and enhanced decision making. In parallel, Tahboub and Dal Yılmaz [41] and Weheida, Ahmed, and Sabaan [42] underlined the importance of theoretical instruction to build knowledge, alongside practical instruction for clinical competency. They recommended reinforcing skills at least every two years. Coll-Badell, Jiménez-Herrera, and Llaurado-Serra [43] similarly advised continuous ECG-monitoring education every five years. Nevertheless, many nursing programs in Korea lack specialized content for high-acuity units, relying on large-group lectures [44]. Yi et al. [45] noted that only 22.7% of 185 programs covered critical care nursing, while emergency nursing was taught in 60.4% of 81 schools [46]. Consequently, many Korean ER and ICU nurses begin practice with limited confidence and competence [47]. Yeom et al. [29] further showed that even nursing students with minimal clinical experience can bolster their performance confidence through experiential learning. As such, educational initiatives should integrate real-life emergency scenarios and merge foundational knowledge with advanced clinical skills to strengthen ECG-monitoring competence. Undergraduate curricula should also embed modules on emergency and critical care to prepare nurses before they enter high-acuity settings.

Our findings revealed that 95.4% of the nurses acknowledged the need for ECG monitoring education, with 90.2% expressing willingness to participate, suggesting that the existing clinical education may not fully address nurses’ educational needs [48]. Online learning (38.7%) and simulation-based methods were particularly favored, likely due to their flexibility for self-directed, repetitive practice [44,49]. In addition, real-time interactive approaches—such as standardized patient sessions and virtual or augmented reality—have demonstrated effectiveness in elevating clinical skills and confidence [50]. These observations underscore the need to enhance ECG monitoring education to satisfy the practical demands of ER and ICU nurses.

Frequent evidence searching also significantly influenced performance confidence. High-pressure environments like ER and ICU require nurses to continuously update their knowledge, but time constraints often hinder comprehensive searches [38,51]. Nurse managers and administrators should provide user-friendly resources (e.g., dedicated terminals and succinct research briefs) to facilitate evidence-based decision making [52]. Strategically deploying well-trained staff—including evidence-based practice mentors or nurse practitioners—can further support the integration of the current guidelines, particularly in ECG monitoring. Moreover, fostering a collaborative workplace culture encourages open communication between nurses and physicians, potentially improving patient outcomes.

Additionally, nurses’ department-specific experience plays a pivotal role in building ECG-monitoring confidence. Those who frequently cared for cardiac patients reported higher assurance, consistent with studies [26,32,38,44,53] indicating that greater exposure to department-specific tasks leads to stronger perceived competency. Optimizing work environments and reducing turnover may enable nurses to gain deeper practical experience, enhancing their confidence and professional commitment [53]. Future research should explore how varying levels of clinical exposure across diverse nursing specialties affect ECG-monitoring proficiency and guide the design of targeted educational interventions.

This study has certain limitations. First, it was conducted in a single tertiary hospital and relied on convenience sampling rather than random selection, which may limit the generalizability of its findings. Therefore, future investigations should employ multi-center or random sampling to better represent the diverse population of ER and ICU nurses. Second, female nurses accounted for about 94.1% of the sample, potentially underrepresenting male perspectives. Although this ratio aligns with the broader Korean nursing demographics [54], subsequent research should strive for more balanced or specifically male-focused samples to assess gender-based differences. Third, reliance on self-reported questionnaires can result in response biases. Incorporating objective observations or simulation-based assessments may yield a more accurate appraisal of nurses’ actual performance confidence. Fourth, the cross-sectional design precludes causal inferences about relationships among variables. Employing longitudinal or experimental studies could clarify how ECG monitoring education impacts both nurse competencies and clinical outcomes over time. Fifth, while the instruments primarily measure theoretical knowledge and self-efficacy, they may not fully capture practical performance. Implementing standardized skill checklists or objective structured clinical examinations (OSCEs) could help reveal discrepancies between theoretical knowledge, self-reported confidence, and real-world practice.

Despite these constraints, the study provides valuable insights into ER and ICU nurses’ ECG monitoring confidence, highlighting the importance of specialized education, evidence updates, and relevant clinical experience. By identifying the factors that enhance performance confidence and uncovering educational needs, we offer foundational data to inform future tailored educational programs aimed at boosting ECG-monitoring competency. Long-term evaluations are warranted to determine the sustained impacts of such programs on clinical outcomes, particularly in specialized, high-acuity environments where evidence-based practices are crucial.

## 5. Conclusions

This study provides foundational data for developing multifaceted educational interventions aimed at strengthening ECG monitoring competency among ER and ICU nurses. By identifying priority topics—based on nurses’ perceived importance and urgency—a systematic and regular education strategy can be designed to ensure continuous access to up-to-date evidence and the provision of targeted training. Supporting such interventions with tailored educational programs and fostering a supportive work environment enables nurses to enhance both knowledge and confidence in ECG monitoring, thereby contributing to improved patient outcomes. Given that the findings highlight factors such as ECG-related education, frequent evidence searching, and clinical experience as the key drivers of performance confidence, future research should examine how various educational interventions—including simulation-based learning, online modules, and advanced clinical practicums—affect nurses’ confidence and actual clinical performance. Longitudinal assessments are essential to determine the sustained efficacy of these programs, guiding further refinements in curriculum design and implementation. Ultimately, a well-structured and evidence-based approach to ECG monitoring education holds the potential to optimize patient care in high-acuity environments such as ER and ICU.

## Figures and Tables

**Figure 1 healthcare-13-00277-f001:**
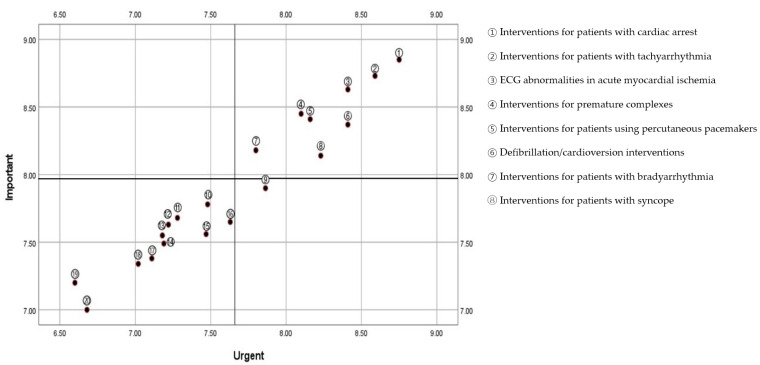
Prioritized ECG monitoring educational content areas using the Eisenhower matrix.

**Table 1 healthcare-13-00277-t001:** General and clinical characteristics of study participants.

Characteristics	Category	*N*	%	Mean ± SD
Gender	Women	144	94.1	
	Men	9	5.9	
Age (years)	20–25	40	26.1	29.74 ± 6.28
	26–30	63	41.2	
	31–35	28	18.3	
	≥36	22	14.4	
Working department	Emergency room	40	26.1	
	Emergency intensive care unit	23	15.1	
	Medical intensive care unit	21	13.7	
	Neurological care unit	34	22.2	
	Surgical intensive care unit	35	22.9	
Education	Associate degree	6	3.9	
	Bachelor degree	129	84.3	
	≥Graduate school	18	11.8	
Professional position	Staff nurse	145	94.8	
	Charge nurse	8	5.2	
Work type	3-shift	143	93.5	
	Regular (non-shift)	10	6.5	
Total clinical experience	<3 years	40	26.1	78.32 ± 75.05
	≥3 years	113	73.9	(months)
Clinical experience within	<3 years	60	39.2	38.37 ± 28.44
current department	≥3 years	93	60.8	(months)
Average number of cardiac				6.39 ± 7.92
patients per week				
Predominant source of evidence	Senior nurses	95	59.5	
for ECG monitoring	Fellow nurses	83	54.2	
	Physicians	54	35.3	
	Textbooks	37	24.2	
	Academic journals or thesis	13	8.5	
	Conference or seminar materials	8	5.2	
	Librarian	5	3.3	
Frequency of ECG monitoring	Less than once per month	66	43.1	
-related evidence search	At least once per month	87	56.9	
Database used for evidence	General web search engines	120	78.4	
Searches over the past month ^a^	Ward manual book	83	54.2	
	Medline/PubMed	43	28.1	
	CINAHL	28	18.3	
	Cochrane library	24	15.7	
Attendance of ECG training	Yes	64	41.8	
sessions/educational programs	No	89	58.2	
Most recent ECG training	<2 years	29	45.3	
Completion (*n* = 64)	≥2 years	35	54.7	
Possession of relevant	Yes	111	72.5	
certification	No	42	27.5	
Attendance at related academic	Yes	1	0.7	
Conference in the past year	No	152	99.3	

Note: SD = standard deviation; ECG = electrocardiographic; ^a^ = duplicates permitted for this item.

**Table 2 healthcare-13-00277-t002:** ECG monitoring knowledge and performance confidence.

Variables	Categories	Mean	SD	Possible Range
Knowledge for ECG monitoring	Overall	13.01	3.08	0~20
Sub-components	Normal waveforms	2.86	1.07	0~4
	Abnormal waveforms	5.60	1.66	0~9
	Visual interpretation of ECG waveforms	4.54	1.29	0~7
Confidence in performing ECG monitoring	Overall	63.47	15.09	0~100
Sub-components	Understanding basic EGC concepts	37.14	9.34	0~60
	Applying ECG monitoring in clinical practice	26.32	6.69	0~40

**Table 3 healthcare-13-00277-t003:** Differences in confidence in performing ECG monitoring according to subjects’ characteristics.

Characteristics	Category	*N*	Self-Confidence					
			Understanding the ECG		Applying ECG Monitoring to Nursing Practice		Overall	
			Mean ± SD	*T* or F (*p*)	Mean ± SD	*T* or F (*p*)	Mean ± SD	*T* or F (*p*)
Gender	Women	144 (94.1)	36.96 ± 9.41	−0.98 (0.328)	26.25 ± 3.79	−0.57 (0.572)	63.21 ± 15.21	−0.96
	Men	9 (5.9)	40.00 ± 8.25		27.59 ± 4.93		67.67 ± 12.90	(0.392)
Age (years)	20–25	40 (26.1)	36.08 ± 10.42	1.18	23.98 ± 7.32	3.05	60.05 ± 17.31	1.80
	26–30	63 (41.2)	36.81 ± 8.03	(0.320)	27.25 ± 6.24	(0.030)	64.06 ± 12.88	(0.150)
	31–35	28 (18.3)	40.07 ± 8.57		28.25 ± 4.98		68.32 ± 12.64	
	≥36	22 (14.4)	36.32 ± 11.47		25.50 ± 7.67		61.82 ± 18.40	
Working department	ER	40 (26.1)	35.48 ± 9.49	3.10 (0.018)	24.60 ± 7.30	1.79 (0.134)	60.08 ± 15.50	2.85
	EICU	23 (15.1)	35.87 ± 10.17		26.70 ± 5.72		62.57 ± 15.34	(0.026)
	MICU	21 (13.7)	32.90 ± 9.72		24.43 ± 7.01		57.33 ± 15.62	
	NCU	34 (22.2)	39.59 ± 8.72		27.41 ± 6.84		67.00 ± 14.90	
	SICU	35 (22.9)	40.06 ± 7.82		28.14 ± 5.77		68.20 ± 12.57	
Education	Associate~Bachelor	135 (88.2)	36.92 ± 8.84	−0.62 (0.544)	26.38 ± 6.52	−0.27 (0.785)	63.32 ± 7.88	1.03 (0.361)
	≥Graduate	18 (11.8)	38.83 ± 12.73		26.83 ± 7.16		65.67 ± 19.12	
Professional position	Staff nurse	145 (94.8)	36.99 ± 9.43	−0.85 (0.398)	26.20 ± 6.65	−1.00	63.19 ± 15.10	−0.97
	Charge nurse	8 (5.2)	39.88 ± 7.85		27.63 ± 7.41	(0.320)	68.50 ± 14.81	(0.334)
Work type	3-shift	143 (93.5)	36.80 ± 9.34	−1.71 (0.089)	26.15 ± 6.69	−1.26	62.95 ± 15.06	−1.62
	Regular	10 (6.5)	42.00 ± 8.51		28.90 ± 6.57	(0.210)	70.90 ± 14.25	(0.107)
Total clinical	<3 years	40 (26.1)	35.95 ± 10.33	−0.94 (0.349)	23.97 ± 6.75	−2.77	59.58 ± 16.95	−1.92
experience	≥3 years	113 (73.9)	37.57 ± 8.99		27.28 ± 6.33	(0.006)	64.85 ± 14.20	(0.057)
Clinical experience within the current department	<3 years	60 (39.2)	35.98 ± 9.73	−1.24 (0.219)	24.69 ± 6.61	−2.65 (0.009)	60.43 ± 15.99	−2.02 (0.045)
	≥3 years	93 (60.8)	37.89 ± 9.07		27.54 ± 6.35		65.43 ± 14.22	
Frequency of ECG monitoring-related evidence search	<once a month	66 (43.1)	34.73 ± 8.84	−2.85 (0.005)	25.26 ± 7.14	−1.73 (0.085)	59.98 ± 14.81	−2.53
	≥once a month	87 (56.9)	38.98 ± 9.36		27.14 ± 6.28		66.11 ± 14.84	(0.012)
Attendance of ECG training/education	Yes	64 (41.8)	38.77 ± 9.99	1.83 (0.069)	27.97 ± 6.15	2.62	66.73 ± 15.12	2.30
	No	89 (58.2)	35.98 ± 8.73		25.15 ± 6.84	(0.010)	61.12 ± 14.71	(0.023)
Most recent ECG training Completion (n = 64)	<2 years	29 (45.3)	42.52 ± 8.98	2.89 (0.005)	29.17 ± 5.22	1.44	71.69 ± 13.47	2.48 (0.016)
	≥2 years	35 (54.7)	35.66 ± 9.83		26.97 ± 6.74	(0.156)	62.63 ± 15.35	
Possession of relevant	Yes	111 (72.5)	37.63 ± 9.48	1.05 (0.297)	26.39 ± 6.73	0.18 (0.856)	64.02 ± 15.27	0.73 (0.467)
certification	No	42 (27.5)	35.86 ± 8.97		26.17 ± 6.67		62.02 ± 14.68	
Attendance at related academic conference in the past year	Yes	1 (0.7)	39.75 ± 16.50	−1.61 (0.109)	21.25 ± 8.42	−1.55	61.00 ± 12.23	−1.69
	No	152 (99.3)	37.34 ± 9.10		26.46 ± 6.62	(0.125)	63.81 ± 14.75	(0.094)

**Table 4 healthcare-13-00277-t004:** Correlation between knowledge and self-confidence in performing ECG monitoring.

Variables	Category	Self-Confidence					
		Understanding the ECG		Applying ECG Monitoring to Nursing Practice		Overall	
		r	*p*	r	*p*	r	*p*
Knowledge for ECG monitoring	Normal waveforms	0.24	0.003	0.17	0.040	0.22	0.005
	Abnormal waveforms	0.17	0.042	0.19	0.020	0.19	0.002
	Visual interpretation of ECG waveforms	0.03	0.743	0.09	0.295	0.05	0.505
	Overall	0.18	0.023	0.19	0.016	0.20	0.013

**Table 5 healthcare-13-00277-t005:** Factors affecting self-confidence in performing ECG monitoring.

Variables	Self-Confidence																
	Understanding the ECG						Applying ECG Monitoring to Nursing Practice						Overall				
	B	S.E	β	*t*	*p*		B	S.E	β	*t*	*p*		B	S.E	β	*t*	*p*
(Constant)	27.71	3.31		8.38	<0.001		18.12	2.35		7.71	<0.001		45.83	5.27		8.69	<0.001
Knowledge	0.28	0.24	0.09	1.17	0.245		0.20	0.17	0.09	1.18	0.239		0.47	0.38	0.10	1.26	0.210
Completion of ECG-related education	3.20	1.44	0.17	2.22	0.028		2.96	1.03	0.22	2.89	0.004		6.16	2.30	0.21	2.68	0.008
Frequency of searching for up-to-date evidence	3.61	1.46	0.20	2.47	0.015		1.43	1.04	0.11	1.38	0.171		5.05	2.34	0.17	2.16	0.032
Clinical experience within current department	2.22	1.45	0.12	1.53	0.129		3.17	1.03	0.24	3.06	0.003		5.39	2.32	0.18	2.32	0.022
Working department	1.77	1.66	0.09	1.07	0.288		2.37	1.18	0.16	2.02	0.046		4.14	2.64	0.12	1.57	0.119
	R^2^ = 0.113 adj. R^2^ = 0.082						R^2^ = 0.163, adj.R^2^ = 0.134						R^2^ = 0.145, adj.R^2^ = 0.116				
	F = 3.71, *p* = 0.003						F = 5.69, *p* < 0.001						F = 4.95, *p* < 0.001				
	Durbin–Watson = 1.932						Durbin–Watson = 1.972						Durbin–Watson = 1.917				

Note: S.E = standard error.

## Data Availability

The data presented and/or analyzed during the current study are available from the corresponding author upon reasonable request.

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
