# Peer review of "Factors Influencing Self-Confidence and Educational Needs in Electrocardiographic Monitoring Among Emergency Room and Intensive Care Unit Nurses"

_healthcare, 2025, doi:10.3390/healthcare13030277_

Round 1
Reviewer 1 Report
Comments and Suggestions for Authors
1. Title and abstract:
The title is clear and reflects the main objective of the study.
The abstract is well structured, adequately summarizing the objectives, methods, main results and conclusions.
2. Introduction:
Provides a good context for the relevance of the ECG in ER and ICU nursing practice.
It is suggested that the gaps identified in the review could be stated more objectively, such as an explicit comparison with previous studies on ECG confidence and knowledge.
3. Methods:
Detailed presentation of the study design, inclusion/exclusion criteria and measurement tools.
Ethical approval is adequately highlighted.
4. Results:
Results are presented in an organized manner, with well-formatted tables.
Robust statistical analysis.
5. Discussion:
Connects findings well with existing literature.
The practical implications for nurse training are clear.
It is suggested: further exploration of the link between the results obtained and settings in other realities; a discussion of the methodological limitations in greater depth.
6. Conclusion:
Clear and concise, it reinforces the practical and academic implications.
Suggestions for future studies are suggested, addressing gaps such as the effectiveness of long-term educational interventions.
Reviewer 2 Report
Comments and Suggestions for Authors
Dear Authors,
It was a pleasure to read your manuscript titled "Factors Influencing Self-Confidence and Educational Needs in Electrocardiographic Monitoring Among Emergency Room and Intensive Care Unit Nurses". The article is well-written and structured, providing a clear and detailed insight into a highly relevant topic in nursing practice. I provide my comments below:
Introduction: The authors provide clear context on the importance of electrocardiographic (ECG) monitoring in emergency and intensive care. The objective is clearly defined. Relevant statistics are cited, highlighting the severity of cardiac issues and the need for prompt intervention. However, I think some sentences can be simplified to improve the flow of this section. For example, instead of "the incidence of acute cardiac arrest increased approximately 1.6 times," you could say "the incidence of acute cardiac arrest increased by x%". I also recommend more clearly emphasizing the specific problem addressed by the study, such as the lack of confidence and knowledge in ECG monitoring among ER and ICU nurses.
Methodology: This section is well-structured and provides clear details on the study design, participants, and data collection methods. Validated tools are used to assess knowledge and confidence in ECG monitoring. However, I have some suggestions:
- Although the sample size calculation is mentioned, you could add a brief explanation of why an effect size of 0.15 was chosen and how it relates to previous studies.
- It would also be appropriate to provide more details about the tools used to measure knowledge and confidence in ECG monitoring, including examples of questions or items.
- In the analysis, why were certain statistical methods (such as ANOVA and multiple regression) chosen, and how do they help answer the research questions?
Discussion: The authors relate the study findings well to the existing literature and highlight the importance of continuous education in ECG monitoring. However, I believe the authors should delve deeper into the interpretation of key results. For example, why are ECG-related education and the search for updated evidence significant factors for confidence in ECG monitoring?
Conclusions: The conclusions provide practical recommendations for improving nurse training. As suggestions in specific areas for future research, the authors could include the evaluation of the long-term effectiveness of educational programs on nurses' clinical performance.
Bibliography: This section is well-resolved and adequately supports the research.
I hope this helps.
Reviewer 3 Report
Comments and Suggestions for Authors
Please see attachment.

Round 2
Reviewer 3 Report
Comments and Suggestions for Authors
Thank you for processing the comments. I wish you all the best for your further research.
Comments on the Quality of English LanguageDear Editor,
With regard to English, a native speaker should check it, so I chose the answer: "The English could be improved to more clearly express the research."